# Sound Field Reconstruction Using Prolate Spheroidal Wave Functions and Sparse Regularization

**DOI:** 10.3390/s23198312

**Published:** 2023-10-08

**Authors:** Xuxin Zhang, Jingjun Lou, Shijian Zhu, Jinfang Lu, Ronghua Li

**Affiliations:** 1College of Power Engineering, Naval University of Engineering, 717 Liberation Avenue, Wuhan 430033, China; zhangxuxin@pku.edu.cn (X.Z.);; 2College of Naval Architecture and Ocean Engineering, Naval University of Engineering, 717 Liberation Avenue, Wuhan 430033, China

**Keywords:** near-field acoustic holography, prolate spheroidal wave functions, orthogonal matching pursuit, sparse regularization

## Abstract

Near-field acoustic holography (NAH) based on compressing sensing (CS) theory enables accurate reconstruction of sound fields using a limited number of sampling points. However, the successful implementation of this technique depends on two crucial factors: (1) the appropriate selection or construction of the spatial basis and (2) an effective sparse regularization process. To enhance reconstruction performance for elongated sound sources, this paper proposes a novel sound field reconstruction method that combines prolate spheroidal wave functions (PSWFs) with the orthogonal matching pursuit (OMP) algorithm. In this method, PSWFs serve as a sparse spatial basis for representing the radiated sound field. The sparse coefficients are determined by the OMP algorithm in a linear subspace composed of basic functions that best match the residual error. The OMP algorithm effectively identifies significant components before potentially selecting incorrect ones by setting an appropriate stopping rule. Numerical simulations are conducted using a line-array source model. The results show that the proposed method can accurately reconstruct the sound pressures of the elongated source model using a relatively small number of samplings. In addition, the proposed method exhibits robustness across a wide frequency range, diverse array configurations and various sampling numbers. The experimental results further validate the feasibility and reliability of the proposed method.

## 1. Introduction

Sound field reconstruction plays a crucial role in analyzing sound source characteristics, understanding vibroacoustic coupling mechanisms, optimizing sound radiation patterns and implementing noise control. Near-field acoustic holography (NAH) [1] is an effective technology for reconstructing the sound field in three-dimensional space. By measuring sound pressures in the near field, NAH can localize sound sources and visualize sound fields with a high resolution. Over the past four decades, numerous NAH algorithms with different sound field spatial bases and inverse methods have been proposed. Some notable methods include spatial discrete Fourier transform-based NAH (DFT-NAH) [2,3,4], the inverse boundary element method (IBEM) [5,6,7], statistically optimized near-field acoustic holography (SONAH) [8,9,10], Helmholtz equation least squares (HELS) [11,12,13] and the equivalent source method (ESM) [14,15,16]. In these algorithms, the sound field is approximated by a series of spatial bases, such as wave functions or simple sources, while the weighting coefficients are estimated by the measured data. Then, a sound field transfer matrix is established between the hologram locations and the reconstruction locations. Finally, accurate reconstruction results can be obtained by including a suitable regularization method to reduce the noise influence. However, achieving high resolution with these NAH algorithms relies on a sufficient number of transducers, which often results in high measurement costs and implementation difficulties in engineering.

Compressing sensing (CS) [17,18,19] offers a promising approach for reducing measurement requirements while preserving a high resolution of the reconstruction results. The fundamental concept of CS theory is based on the assumption that the sound field can be sparsely represented using only a few non-zero weighting coefficients. By leveraging this sparsity, accurate reconstruction of the sound field can be achieved using a relatively small number of samplings. The successful integration of CS theory into NAH depends on two key factors: (1) the appropriate selection or construction of the spatial basis and (2) an effective sparse regularization process. When the spatial basis matches the characteristics of the sound field or the sound sources well, the sparsity of the sound field representation can be promoted. As a result, the inverse problem in NAH is simplified, and the accuracy of the reconstruction results is improved. Chardon et al. [19] first combined a set of plane wave functions and *l*_1_-norm sparse regularization to reconstruct the normal velocity of plates in 2012. It was shown that sparsity principles for NAH were valid and the measurement requirements could be mitigated. Furthermore, to address spatially sparse sources, Hald [20] used a set of monopole point sources to match the sound field, while the sparse regularization process was performed by an iterative algorithm called fast wideband holography (WBH). For spatially extended sources, Fernandez-Grande [21] used block-sparse regularization along with ESM to promote the sparse solutions. Bi [22] constructed a sparse basis obtained from the eigen-decomposition of the power resistance matrix. Hu [23] constructed a sparse basis by ESM and the singular value decomposition (SVD). Additionally, plane wave functions [24], cylindrical wave functions and spherical wave functions [25,26] can also be selected as the sparse spatial basis. As for the sparse regularization, various algorithms have been applied to NAH successfully, such as direct *l*_1_-norm sparse regularization [27], WBH [20], iterative sparse methods [28], Bayesian compressive sensing [29,30], alternating direction method of multipliers [31], block sparse Bayesian learning [32,33] and so on. Note that the computational efficiency, effective frequency range and noise robustness of these sparse regularization algorithms may vary substantially with different spatial bases and sound sources [28,34]. Therefore, it is important to consider the characteristics of both the spatial bases and sound sources when implementing sparse regularization algorithms.

Considering elongated sound sources, such as unmanned underwater vehicles with large aspect ratios, line-array speakers, rockets and gas turbines, the radiated sound fields may exhibit complex vertical directivities in three-dimensional space [35]. Cylindrical wave functions are frequently used to represent this kind of sound field. In previous studies, Williams [36,37] utilized cylindrical wave functions within the Fourier transform framework, known as cylindrical NAH, to reconstruct the radiated sound field of finite cylindrical shells. Lee [38] employed the virtual coherence method and cylindrical NAH to visualize the sound field of a small ducted fan assembly. It should be noted that the performance of cylindrical NAH was often affected by spatial windowing effects and wraparound errors. To address this issue, Lee [39] proposed an iterative data extrapolation algorithm to extend the measurement aperture. Cho [40] combined cylindrical wave functions with the SONAH algorithm to reconstruct the sound field of a cylindrical refrigeration compressor. However, when dealing with more complex, elongated sound sources, cylindrical wave functions alone may not be adequate for accurately representing the sound field. Consequently, Williams [4] employed the fast Fourier transform to solve a partial domain problem caused by the unbaffled cylinder with flat endcaps. Cheng [10] expressed the sound field of large conical surfaces as a superposition of characteristic elementary waves, which are constructed by cylindrical wave functions through cylinder-cone coordinates transformation. Overall, it is beneficial to carefully select or construct a spatial basis that aligns well with the elongated sources. Such alignment has the potential to enhance the efficacy of spatial basis in accurately representing the sound field, ultimately leading to precise reconstruction results.

To apply CS theory to sound field reconstruction for elongated sources, the spatial basis should possess an excellent ability to represent the featured sound field. In other words, the spatial basis should satisfy the conditions required for employing CS methods, which imply that the radiated sound field can be sparsely or compressibly represented. The references in this field are few. However, according to our simulation tests for a simple unmanned underwater vehicle model, the reconstruction results near the endcaps do not perform well when using cylindrical wave functions. As a result, prolate spheroidal wave functions (PSWFs) [41,42] are investigated and utilized as a novel sparse spatial basis for reconstructing the radiated sound field of an elongated sound source in our study. In fact, PSWFs are the eigenfunctions of the Helmholtz equation in the prolate spheroidal coordinate system. Similar to plane wave functions, cylindrical wave functions and spherical wave functions, PSWFs exhibit orthogonality and completeness in the spatial domain. Additionally, the wavefronts of PSWFs can be set nearly conformal to the elongated sources, which infers that PSWFs possess a strong capability to accurately represent the radiated sound field. However, the usage of PSWFs has always been limited due to their complex mathematical expressions and the challenges associated with performing precise calculations. In recent times, researchers, including Van Buren [43,44,45], Falloon [46], Ogburn [47] and Adelman [48], have made efforts to enhance the accessibility of PSWFs. They have generously shared their code scripts publicly, which serve as valuable resources for others interested in utilizing PSWFs. These contributions have significantly reduced barriers and facilitated the practical implementation of PSWFs in various applications, including acoustic scattering [49], acoustic radiation [50] and electromagnetic scattering [42].

The effective sparse regularization process also plays a crucial role in applying CS theory to reconstruct sound fields for elongated sources. However, our primary focus in this study is to investigate whether the use of PSWFs can enhance the accuracy of reconstruction results for elongated sources. To this end, we employ the orthogonal matching pursuit (OMP) [51,52] method, which is one of the most reliable CS algorithms for sparse regularization. Specifically, OMP is an iterative algorithm that reconstructs a signal by progressively selecting atoms or features that are highly correlated with the target signal. During each iteration, the algorithm generates an approximation of a given signal, which guarantees two properties: the projection of the signal onto the subspace generated by the selected atoms is orthogonal, and the norm of the corresponding residual error is minimized.

This paper presents a novel method, termed PSWF-OMP, for reconstructing the radiated sound field of an elongated sound source by combining PSWFs with the OMP algorithm. PSWFs are utilized as a sparse spatial basis to accurately represent the sound field, while the OMP algorithm is employed to find a sparse solution by identifying the most relevant basis vectors and minimizing errors under a limited number of measurements. Numerical and experimental evaluations are conducted to assess the performance of the proposed PSWF-OMP method. Furthermore, a comparison is made between PSWF-OMP and the traditional SONAH algorithm to highlight the advantages of the former in dealing with elongated sound sources when measurements are not sufficient. Additionally, PSWF-OMP is compared to a method that combines cylindrical wave representation with the OMP algorithm to demonstrate the compatibility of PSWFs with elongated sound sources. Moreover, the influences of sampling numbers and array configurations are investigated.

The outline of this paper is organized as follows. Section 2 introduces the basic theories of PSWFs and sparse regularization. In Section 3, numerical simulations are performed to validate the proposed method, and the influences of sampling numbers and array configurations are discussed. The feasibility of the proposed method is further demonstrated through experiments in Section 4. Lastly, the conclusions are summarized in Section 5.

## 2. Theory

### 2.1. Sound Field Expansion Using Prolate Spheroidal Wave Functions

Considering a time-harmonic sound field radiated by an elongated sound source in an unbounded homogenous medium, the complex sound pressure p(r) satisfies the Helmholtz equation:(1)$$\left(\nabla^{2}+k^{2}\right)p\left(r\right)=0,$$
where $\nabla^{2}$ is the Laplace operator, r is the position of a field point in the unbounded exterior region, k=ω/v is the wave number, ω is the angular frequency, and v is the sound speed. Since the Helmholtz equation is separable in curvilinear orthogonal systems, p(r) is often expressed as a summation of certain basis wave functions. To effectively address the sound radiation problem involving elongated geometries, the prolate spheroidal coordinate system is employed in this study.

The prolate spheroidal coordinate system [41] is formed by rotating the two-dimensional elliptic coordinate system about the major axis of the ellipses, consisting of confocal ellipses and hyperbolas. In the prolate spheroidal coordinate system, a point is defined by three coordinates (ξ,η,φ). As shown in Figure 1, the prolate spheroids (ξ=const), hyperboloids (η=const) and half-planes (φ=const) are three orthogonal coordinate surfaces. d is the interfocal distance. The major semi-axis of the prolate spheroid is dξ/2, while the minor semi-axis is $d\sqrt{\xi^{2}-1}/2$. Note that, a different d defines different prolate spheroid systems, which can be manipulated to set nearly conformal prolate spheroidal surfaces to elongated sound sources. In general, the *z*-axis is chosen as the axis of rotation, and prolate spheroidal coordinates (ξ,η,φ) are related to Cartesian coordinates (x,y,z) by
(2)$$\left\{\begin{array}{l} x=\frac{d}{2}\sqrt{\left(\xi^{2}-1)(1-\eta^{2}\right)}\cos\varphi ,\\ y=\frac{d}{2}\sqrt{\left(\xi^{2}-1)(1-\eta^{2}\right)}\sin\varphi ,\\ z=\frac{d}{2}\xi \eta , \end{array}\right.$$
where 1≤ξ<∞, −1≤η≤1, and  0≤φ≤2π. Inversely, (ξ,η,φ) can be derived by (x,y,z) as follows:(3)$$\left\{\begin{array}{l} \xi =\left(R_{1}+R_{2}\right)/d,\\ \eta =\left(R_{1}-R_{2}\right)/d,\\ \varphi =\mathrm{atan}2(y/x), \end{array}\right.$$
where $R_{1}=\sqrt{{\left(z+d/2\right)}^{2}+x^{2}+y^{2}}$ and $R_{2}=\sqrt{{\left(z-d/2\right)}^{2}+x^{2}+y^{2}}$ are the distances from a field point to two foci, respectively. atan2(•) represents the four-quadrant inverse tangent function.

In the prolate spheroidal coordinate system, the Helmholtz equation is rewritten as follows:(4)$$\left[\frac{\partial }{\partial \xi }(\xi^{2}-1)\frac{\partial }{\partial \xi }+\frac{\partial }{\partial \eta }(1-\eta^{2})\frac{\partial }{\partial \eta }+\frac{\xi^{2}-\eta^{2}}{\left(\xi^{2}-1)(1-\eta^{2}\right)}\frac{\partial^{2}}{\partial \varphi^{2}}+c^{2}(\xi^{2}-\eta^{2})\right]p(r)=0,$$
where c=kd/2=πd/λ. Because the prolate spheroidal coordinate system is a curvilinear orthogonal system, we can employ the separation of variables to derive three uncoupled ordinary differential equations [41]. The solutions to the Helmholtz equation can be derived as follows:(5)$$R_{mn}^{\left(q\right)}(c,\xi )S_{mn}(c,\eta )\left\{\begin{array}{l} \cos(m\varphi )\\ \sin(m\varphi ) \end{array}\right.,\;\;m=0,1,2,\ldots \;\mathrm{and}\;n=m,m+1,m+2,\ldots$$
where $S_{mn}(c,\eta )$ are the first kind of prolate spheroidal angle functions of degree n and order m, $R_{mn}^{\left(q\right)}(c,\xi )$ are the *q*th kind of prolate spheroidal radial functions with q=1,2,3,4, and cos(mφ) and sin(mφ) are the azimuthal functions.

Similar to spherical Hankel functions, the third and fourth kinds of prolate spheroidal radial functions can be expressed by the first and second kinds [53]:(6)$$\left\{\begin{array}{l} R_{mn}^{\left(3\right)}\left(c,\xi \right)=R_{mn}^{\left(1\right)}\left(c,\xi \right)+iR_{mn}^{\left(2\right)}\left(c,\xi \right),\\ R_{mn}^{\left(4\right)}\left(c,\xi \right)=R_{mn}^{\left(1\right)}\left(c,\xi \right)-iR_{mn}^{\left(2\right)}\left(c,\xi \right), \end{array}\right.$$
where i represents the imaginary unit. When the time dependency is assumed as $e^{-i\omega t}$, it is reasonable to regard $R_{mn}^{\left(3\right)}\left(c,\xi \right)$ and $R_{mn}^{\left(4\right)}\left(c,\xi \right)$ as the distance dependency of the outgoing and incoming wave functions, respectively [54]. Analogous to spherical harmonics, prolate spheroidal harmonics can be generated by multiplying prolate spheroidal angle functions and azimuthal functions [47]. In this case, the azimuthal functions should be changed from cos(mφ) and sin(mφ) to $e^{im\varphi }$, and the negative integers of m must be considered. As a result, the definitions of prolate spheroidal radial functions and angle functions are extended:(7)$$R_{-mn}^{\left(3\right)}(c,\xi )=R_{mn}^{\left(3\right)}(c,\xi ),$$
(8)$$R_{-mn}^{\left(4\right)}(c,\xi )=R_{mn}^{\left(4\right)}(c,\xi ),$$
(9)$$S_{-mn}(c,\eta )={\left(-1\right)}^{m}\frac{(n-m)!}{(n+m)!}S_{mn}(c,\eta ).$$

In the following, the subscripts of the three functions are redefined as n=0,1,…,∞ and m=−n,…,−1,0,1,…,n. Thus, prolate spheroidal harmonics can be constructed as follows:(10)$$H_{n}^{m}(\eta ,\varphi ;c)=\sqrt{\frac{2n+1}{4\pi }\frac{(n-m)!}{(n+m)!}}S_{mn}(c,\eta )e^{im\varphi }.$$

It should be noted that $H_{n}^{m}(\eta ,\varphi ;c)$ are normalized and possess a unique double orthogonality property [47]:(11)$$\int_{0}^{2\pi }d\varphi \int_{-1}^{1}H_{n}^{m}(\eta ,\varphi ;c)H_{n^{\prime }}^{m^{\prime }}{\left(\eta ,\varphi ;c\right)}^{*}d\eta =\delta_{mm^{\prime }}\delta_{nn^{\prime }},$$
where $\delta_{mm^{\prime }}$ and $\delta_{nn^{\prime }}$ are the Kronecker delta functions, and “*” represents the complex conjugation. The double orthogonality property implies that $H_{n}^{m}(\eta ,\varphi ;c)$ are orthogonal with respect to a unit weighting function rather than the Jacobian matrix. Though the double orthogonality property does not involve a conventional surface area integral, it allows for simplifications in computations and leads to efficient mathematical formulations. As a result, prolate spheroidal harmonics can form an orthogonal basis over the product domain [47]. The real parts of the first few outgoing prolate spheroidal harmonics for positive *m* are demonstrated in Figure 2.

From the properties discussed, the outgoing and incoming PSWFs can be constructed by combining prolate spheroidal radial functions with prolate spheroidal harmonics:(12)$$\Phi_{mn}^{out}(\xi ,\eta ,\varphi ;c)=R_{mn}^{\left(3\right)}(c,\xi )H_{n}^{m}(\eta ,\varphi ;c),$$
(13)$$\Phi_{mn}^{in}(\xi ,\eta ,\varphi ;c)=R_{mn}^{\left(4\right)}(c,\xi )H_{n}^{m}(\eta ,\varphi ;c).$$

It is inferred that outgoing and incoming PSWFs can form an orthogonal and complete spatial basis. This implies that any radiated sound field can be represented by outgoing and incoming PSWFs as follows:(14)$$p(\xi ,\eta ,\varphi )=\sum_{n=0}^{\infty }\sum_{m=-n}^{n}\left[a_{mn}(c)\Phi_{mn}^{out}(\xi ,\eta ,\varphi ;c)+b_{mn}(c)\Phi_{mn}^{in}(\xi ,\eta ,\varphi ;c)\right],$$
where $a_{mn}(c)$ and $b_{mn}(c)$ are complex weighting coefficients. For an exterior acoustic problem, there are no incoming PSWFs converging inward, so $b_{mn}(c)=0$. Therefore, the exact solution to an exterior problem can be expressed as follows:(15)$$p(\xi ,\eta ,\varphi )=\sum_{n=0}^{\infty }\sum_{m=-n}^{n}a_{mn}(c)\Phi_{mn}^{out}(\xi ,\eta ,\varphi ;c).$$

However, in practice, this expansion must be truncated to a finite order for computational purposes:(16)$$p(\xi ,\eta ,\varphi )\approx \sum_{j=1}^{J}a_{j}(c)\Phi_{j}^{out}(\xi ,\eta ,\varphi ;c),$$
where the subscripts of the outgoing PSWFs are replaced by $j=n^{2}+n+m+1$.

### 2.2. Sound Field Reconstruction Model

The sound field reconstruction model involves the measurement of sound pressures in the near field of sound sources and the subsequent reconstruction of the three-dimensional sound field using measured data as boundary conditions. Assuming that there are *L* transducers on the hologram plane and the truncation number *J* is appropriately chosen, the measured sound pressures can be expressed as follows:(17)$$p_{h}=\Psi_{h}a+n,$$
where $p_{h}={\left[p(r_{h1}),p(r_{h2}),\ldots ,p(r_{hl}),\ldots p(r_{hL})\right]}^{T}$ is a column vector representing the measured pressures, $a={\left[a_{1}(c),a_{2}(c),\ldots ,a_{j}(c),\ldots a_{J}(c)\right]}^{T}$ is a column vector of the weighting coefficients, n is the white Gaussian noise vector, and $\Psi_{h}$ is an L×J matrix of the outgoing PSWFs evaluated at the measurement points:
(18)$$\Psi_{h}=[\begin{array}{cccccc} \Phi_{1}^{\mathrm{out}}\left(r_{h1};c\right) & \Phi_{2}^{\mathrm{out}}\left(r_{h1};c\right) & \cdots & \Phi_{j}^{\mathrm{out}}\left(r_{h1};c\right) & \cdots & \Phi_{J}^{\mathrm{out}}\left(r_{h1};c\right)\\ \Phi_{1}^{\mathrm{out}}\left(r_{h2};c\right) & \Phi_{2}^{\mathrm{out}}\left(r_{h2};c\right) & \cdots & \Phi_{j}^{\mathrm{out}}\left(r_{h2};c\right) & \cdots & \Phi_{J}^{\mathrm{out}}\left(r_{h2};c\right)\\ \vdots & \vdots & \ddots & \vdots & \ddots & \vdots \\ \Phi_{1}^{\mathrm{out}}\left(r_{hl};c\right) & \Phi_{2}^{\mathrm{out}}\left(r_{hl};c\right) & \cdots & \Phi_{j}^{\mathrm{out}}\left(r_{hl};c\right) & \cdots & \Phi_{J}^{\mathrm{out}}\left(r_{hl};c\right)\\ \vdots & \vdots & \ddots & \vdots & \ddots & \vdots \\ \Phi_{1}^{\mathrm{out}}\left(r_{hL};c\right) & \Phi_{2}^{\mathrm{out}}\left(r_{hL};c\right) & \cdots & \Phi_{j}^{\mathrm{out}}\left(r_{hL};c\right) & \cdots & \Phi_{J}^{\mathrm{out}}\left(r_{hL};c\right) \end{array}]$$

Once the weighting coefficients are determined, the pressures on the reconstruction plane can be expressed as follows:(19)$$p_{r}=\Psi_{r}a,$$
where $p_{r}$ is the column vector of reconstructed pressures and $\Psi_{r}$ is a matrix of outgoing PSWFs evaluated at reconstruction locations.

The determination of the weighting coefficients is an acoustic inverse problem. Since the matrix $\Psi_{h}$ in Equation (17) often performs ill-conditioned, the measurement noise can be greatly amplified during the inverse-solving process, resulting in compromised outcomes. To deal with the ill-posed problem, Tikhonov regularization [55] is usually used in conventional NAH:(20)$$\arg\underset{a}{\min}\left\{{\left\|p_{h}-\Psi_{h}a\right\|}_{2}^{2}+\lambda {\left\|a\right\|}_{2}^{2}\right\},$$
where λ is the regularization parameter. By minimizing the *l*_2_-norm of the residual vector and the coefficient vector, the least-square solution can be found:(21)$$a={\left(\Psi_{h}^{H}\Psi_{h}+\lambda I\right)}^{-1}\Psi_{h}^{H}p_{h}$$
where I is the identity diagonal matrix, “H” represents the Hermitian transpose and ${\left(•\right)}^{-1}$ represents the pseudo-inverse. Generally, λ can be determined via the generalized cross validation (GCV) method or the L-curve method [55]. However, the least-square coefficient vector often performs “smooth” instead of “sparse”, which may allocate too much energy to high-wavenumber components and amplify the impact of noise contamination. In ESM, the least-square solution may lead to ghost sources near the real sound source and decrease the accuracy of sound source recognition. Furthermore, in order to guarantee the uniqueness of the least-square solution, Equation (17) should remain overdetermined. This implies that the number of measurements needs to exceed the number of wave functions.

### 2.3. Sparse Regularization via Orthogonal Matching Pursuit

In engineering applications, sufficient measurements cannot always be satisfied. When the number of measurements is fewer than the number of wave functions, the inverse-solving problem becomes underdetermined. In this case, sparse regularization is used to enhance the accuracy and reliability of the sound field reconstruction. The fundamental principle behind sparse regularization is that the real sound field exhibits sparsity and can be accurately represented using only a few significant coefficients in a specific transform domain. In our study, PSWFs are specially chosen for elongated sources as the sparse spatial basis to fulfill the sparsity condition. The use of PSWFs offers advantages in three main aspects. Firstly, the completeness of PSWFs allows for the reconstruction of any type of radiated sound field. Secondly, the orthogonality of PSWFs enhances the uniqueness of the weighting coefficients during the inverse-solving process. Thirdly, the nearly conformal wavefronts of PSWFs contribute to a more concise representation of the sound field, resulting in increased sparsity.

When the sparse basis is determined, the weighting coefficients are usually obtained by solving the *l*_1_-norm minimization problem:(22)$$\arg\underset{a}{\min}{\left\|a\right\|}_{1}\;\mathrm{subject}\;\mathrm{to}\;{\left\|p_{h}-\Psi_{h}a\right\|}_{2}\leq \varepsilon ,$$
where ε is the error threshold. In this study, sparse regularization is performed by the OMP algorithm. As an iterative and greedy algorithm, the OMP has been widely used due to its simplicity and competitive performance. In fact, OMP can incrementally identify the significant features that contribute the most to the signal of interest. At each step, OMP selects a sparse basis function that is most correlated with the current residual. The selected sparse basis function is then added to a linear subspace composed of previously selected basis functions. Finally, the residual is updated by projecting the measured pressures onto the linear subspace. Note that, the basis functions are called “atoms” in OMP, and the linear subspace is called “sub-dictionary”. In addition, the residual after each step in the OMP algorithm is orthogonal to all the selected atoms, so no atom is selected twice, and the set of selected atoms grows at each step.

Since the OMP algorithm is usually applied to problems involving real-valued data, the complex vector $p_{h}$ and matrix $\Psi_{h}$ should be separated into real and imaginary parts. Therefore, the complex-valued linear system in Equation (17) is transformed into a real-valued linear system as follows:(23)$${\tilde{p}}_{h}={\tilde{\Psi }}_{h}\tilde{a}+\tilde{n},$$
where
(24)p˜h=RephImph,a˜=ReaIma,n˜=RenImn,
and
(25)Ψ˜h=ReΨh−ImΨhImΨhReΨh.

Re(·) and Im· represent the real and imaginary parts, respectively. Furthermore, the framework of the OMP algorithm is presented in Algorithm 1. Note that the selection of the most correlated basis function with the residual is accomplished by finding the largest inner product. The index of the selected basis function is then recorded in $\Lambda_{k}$. Simultaneously, the weighting coefficient vector in the *k*th iteration is estimated using a least square approach.
**Algorithm 1** OMP Algorithm.1: $\mathrm{Input}\;\mathrm{matrix}\;{\tilde{\Psi }}_{h},\;\mathrm{column}\;\mathrm{vector}\;{\tilde{p}}_{h}$ and error threshold *ε*.2: **Initialize** residual vector $R_{0}={\tilde{p}}_{h}$, linear subspace $\Omega_{0}=\emptyset$, indices of selected basis    functions $\Lambda_{0}=\emptyset \;\mathrm{and}\;\mathrm{iteration}\;\mathrm{count}\;k=0.$3: 
$\mathrm{While}\;{\left\|R_{k}\right\|}_{2}>\varepsilon$4:   
k++5:   
$index=\arg\underset{j\notin \Lambda_{k-1}}{\max}\left|R_{k-1}^{H}{\tilde{\Psi }}_{h}\right|$6:   
$\Lambda_{k}=\Lambda_{k-1}\cup \left\{index\right\}$7:   
$\Omega_{k}=\Omega_{k-1}\cup {\left\{{\tilde{\Psi }}_{h}\right\}}_{index}$8:   
${\tilde{a}}_{k}^{sub}={\left(\Omega_{k}^{H}\Omega_{k}\right)}^{-1}\Omega_{k}^{H}{\tilde{p}}_{h}$9:   
$R_{k}={\tilde{p}}_{h}-\Omega_{k}{\tilde{a}}_{k}^{sub}$10:   **End while**11:   **Return**
$\Lambda_{k}\;{\tilde{a}}_{k}^{sub}\;R_{k}$

One of the key components of OMP is the stopping rule. As Cai [52] suggested, if the noise vector is bounded with ${\left\|n\right\|}_{2}\leq \varepsilon$, the stopping rule can be set to ${\left\|R_{k}\right\|}_{2}\leq \varepsilon$. In this case, OMP can identify all the large components before possibly selecting incorrect ones. Furthermore, OMP can demonstrate good tolerance to additive noise in the measurements. Due to its iterative nature and the ability to selectively choose basis functions that contribute the most to reducing the residual error, unnecessary or redundant information can be suppressed, resulting in a more concise and accurate representation of the sound field. In other words, the OMP algorithm can mitigate noise by highlighting its essential components while filtering out irrelevant details.

## 3. Numerical Simulations

In the simulation, the proposed method is evaluated by a set of line-array monopoles. As shown in Figure 3, 101 monopoles are placed along the *z*-axis, evenly distributed in the range of z=−0.5~0.5 m. The monopoles are spaced 0.01 m from each other. Additionally, all the monopoles have identical source strengths and phases. In such a configuration, a strong interference effect can be stimulated, which means that the radiated sound field exhibits a strong near-field effect at low frequencies and complex vertical directivity at high frequencies. The origin of the Cartesian coordinate system is set at the center of the elongated source model. The fluid medium is assumed to be air. The sound speed is set to 340 m/s. The theoretical sound pressures from the line-array monopoles are all calculated with the free-field Green’s function. The hologram surface is located at x=0.2 m with a size of $1.2\times 1.2{\;m}^{2}$. Simulated pressures are measured on the hologram surface with a total of 25×25=625 evenly distributed sampling points. To evaluate the proposed method in a sparse framework, a subset of sampling points is randomly selected from the total 625 measurements. Additional white Gaussian noise with a signal-to-noise ratio (SNR) of 30 dB is added to the pressures on the hologram surface to simulate the real environment. Meanwhile, the reconstruction surface is the same size as the hologram surface and is placed at x=0.1 m. Similar to the hologram surface, the reconstruction surface is also meshed into 25×25=625 points with a grid spacing of 0.05 m.

The performance of the proposed method is compared with traditional SONAH algorithms [9] that employ cylindrical wave functions to represent the radiated sound field of elongated sources. A modified Tikhonove regularization method is employed in the SONAH algorithm, while the regular parameter is found through the GCV method. Additionally, a sound field reconstruction method that combines cylindrical wave representation with the OMP algorithm is compared with the proposed method to reveal whether PSWFs can match with elongated sound sources well. It is important to note that both PSWFs and cylindrical wave functions are orthogonal and complete in three-dimensional space, allowing them to approximate any type of sound field. However, there are distinct differences in the characteristics of wavefronts and spatial symmetry between these two wave functions. As a result, the conciseness of representation between these two wave functions may vary when depicting the sound field. Considering the noise influence and the ill-posed problem in NAH, a sparse and accurate representation of the sound field is always being pursued. Consequently, the ability of PSWFs and cylindrical wave functions are investigated and compared with respect to the line-array source model. In the simulation, cylindrical wave functions are formulated as follows:(26)$$\Phi_{n,k_{z}}^{c}(\rho ,\varphi ,z)=\frac{{\overset{⌢}{H}}_{n}^{\left(1\right)}\left(k_{r}\rho \right)}{{\overset{⌢}{H}}_{n}^{\left(1\right)}\left(k_{r}\rho_{0}\right)}e^{in\varphi }e^{ik_{z}z},$$
(27)$$k_{r}=\left\{\begin{array}{l} \sqrt{k^{2}-k_{z}^{2}},k^{2}>k_{z}^{2}\\ i\sqrt{k_{z}^{2}-k^{2}},k^{2}<k_{z}^{2} \end{array}\right.,$$
where $\Phi_{n,k_{z}}^{c}(\rho ,\varphi ,z)$ are outgoing cylindrical wave functions with respect to cylindrical coordinates (ρ,φ,z); ${\overset{⌢}{H}}_{n}^{\left(1\right)}\left(k_{r}\rho \right)$ are the *n*th-order Hankel functions of the first kind with n=−9,−8,…,−1,0,1,…,8,9; $k_{r}$ is the radial wavenumber; $\rho_{0}$ is a small reference radius; ${\left|k_{z}\right|}_{\max}=\pi /0.1{\;m}^{-1}$; and $\Delta k_{z}=2\pi /2.4{\;m}^{-1}$. Since the maximum analytical frequency is 1000 Hz, corresponding to the wavelength of 0.34 m, the configurations of the cylindrical wave functions can meet our requirements to represent the sound field [1]. For the sake of simplicity, the method combining cylindrical wave functions with the OMP algorithm is abbreviated as CWF-OMP. Traditional SONAH algorithms that employ cylindrical wave functions are abbreviated as CWF-SONAH. As for PSWFs, the interfocal distance is set to d=1.2 m, which is slightly larger than the length of the elongated source model. In this case, as the singularities in the three-dimensional sound field, all the sound sources are incorporated into the singularities of the PSWFs (the line between two focuses), making the sound field reconstruction reliable and robust. Meanwhile, the wavefronts of the PSWFs can be set nearly conformal to the sound sources, promoting the sparse representation of the radiated sound field. Note that, the error threshold is set to $\varepsilon =n_{2}$ in the OMP algorithm.

To quantitatively evaluate the reconstruction performance, the reconstruction error is defined as follows:(28)$$RE=\frac{{\left\|p_{cal}-p_{the}\right\|}_{2}}{{\left\|p_{the}\right\|}_{2}}\times 100\%.$$
where $p_{cal}$ is the reconstructed pressure and $p_{the}$ is the theoretical pressure.

### 3.1. Validation of the Proposed Method with Sparse Sampling

In this subsection, the sound field reconstruction is carried out at 500 Hz and 1000 Hz based on a small number of samplings. As shown in Figure 4, the number of randomly selected sampling points is set to 50, which is fewer than one tenth of the total number of sampling points (625) on the hologram surface. Figure 5 compares the reconstructed pressures by CWF-SONAH, CWF-OMP and PSWF-OMP with the theoretical pressures at 500 Hz. Figure 6 compares the reconstructed pressures at 1000 Hz. It is apparent that the theoretical pressures exhibit long strip patterns along the *z*-axis at both 500 Hz and 1000 Hz. The interference effect is particularly pronounced at 1000 Hz, where three vertical lobes can be observed in Figure 6a. Note that the sound field reconstruction results obtained through PSWF-OMP demonstrate a good match with the theoretical pressures, as depicted in Figure 5d and Figure 6d. The reconstruction errors are evaluated to be 5.5% at 500 Hz and 4.5% at 1000 Hz, indicating a high level of accuracy. In contrast, the reconstruction errors of CWF-OMP are found to be 13.7% at 500 Hz and 17.2% at 1000 Hz, both of which are higher compared to PSWF-OMP. The reconstruction errors of CWF-SONAH are 14.6% at 500 Hz and 4.7% at 1000 Hz. The performance of CWF-SONAH at low frequencies appears to be inadequate, indicating the need for further measurements. Additionally, the reconstructed patterns obtained through CWF-OMP exhibit compression in the *z*-direction at both frequencies, suggesting a potential occurrence of larger errors near the edge of the reconstruction surface. The reconstructed pattern obtained through CWF-SONAH at 500 Hz manifests obvious distortion, which may be caused by incorrect selection of the regularization parameter. Furthermore, in terms of peak distribution features, PSWF-OMP demonstrates a closer visual resemblance compared to CWF-OMP at both frequencies. Overall, the results demonstrate that PSWF-OMP yields good reconstruction outcomes at both low and high frequencies. The reconstruction errors obtained through the proposed method are all below 6%, indicating its effectiveness in achieving sound field reconstruction with a reduced number of measurement points.

In order to gain a more detailed observation of the reconstruction results, Figure 7 displays the sound pressure levels (SPLs) calculated through CWF-SONAH, CWF-OMP and PSWF-OMP at 500 Hz and 1000 Hz along the middle column line and middle row line on the reconstruction surface. In Figure 7a,c, noticeable disparities between the reconstructed SPLs by CWF-OMP and the theoretical values can be observed at edge points. For CWF-SONAH, the evident deviation in Figure 7a suggests a failure in sound field reconstruction at 500 Hz. Additionally, Figure 7d reveals the presence of redundant ripples obtained through CWF-OMP at edge points, which could potentially stem from incorporating high wavenumber components. This suggests that a few high-order cylindrical wave functions have been utilized to approximate the interference pattern on the hologram surface. However, due to measurement noise, it becomes challenging to determine the accurate weighting coefficients. As a result, a portion of the energy that belongs to the lower wavenumber components is misallocated to the higher wavenumber components. During the inverse reconstruction process, the amplitudes of these high-order wave functions undergo significant amplification, leading to the emergence of ripples on the reconstruction surface, and consequently resulting in substantial reconstruction errors. In contrast, the PSWF-OMP method demonstrates a remarkable ability to achieve close agreement with the theoretical values along both the middle column line and middle row line. The accurate reconstruction at edge points indicates the strong abilities of PSWFs in representing the radiated sound field of line-array monopoles.

Figure 8a–d illustrates the normalized coefficient amplitudes obtained through CWF-SONAH, CWF-OMP and PSWF-OMP at 500 Hz and 1000 Hz. The values represented on the horizontal axes correspond to the wavenumber components, with larger absolute values indicating higher wavenumbers. It can be seen that at 500 Hz, CWF-OMP employs eight wave functions to approximate the radiated sound field, whereas PSWF-OMP only utilizes two wave functions. Similarly, at 1000 Hz, CWF-OMP employs 11 wave functions, while PSWF-OMP employs only 3 wave functions. As for CWF-SONAH, more wave functions are always used to represent the sound field at both frequencies. This discrepancy in the number of wave functions employed by the three methods suggests that PSWF-OMP is able to achieve comparable or even better reconstruction accuracy with fewer wave functions. The concise representation in PSWF-OMP highlights the efficiency and effectiveness of the PSWF-OMP approach in reconstructing the sound field. It can be seen that both CWF-OMP and PSWF-OMP preferentially choose low-order wave functions for modeling the radiated sound field. However, PSWF-OMP can realize more accurate reconstruction results (see Figure 6 and Figure 7). This characteristic signifies the strong matching performance of PSWFs to the elongated sound source model. Furthermore, it is important to highlight that a considerable portion of the weighting coefficients, specifically those associated with high wavenumber components, are set to zero in PSWF-OMP. This can efficiently mitigate the influences of noise. Overall, PSWF-OMP performs a sparser representation, and the impact of noise on the reconstructed pressures can be greatly reduced, leading to improved accuracy and quality of the reconstruction results.

### 3.2. Effect of Array Configurations

To examine the influence of sampling array randomness on the reconstruction results, a total of 30 different sampling array configurations are utilized. Each configuration consists of 50 sampling points randomly selected from the original 625 measurements. The obtained results, as shown in Figure 9, reveal that the mean reconstruction error achieved by PSWF-OMP is 5.5% at 500 Hz and 4.3% at 1000 Hz. In contrast, the mean reconstruction error attained by CWF-OMP is significantly higher, reaching 13.7% at 500 Hz and 13.5% at 1000 Hz. The mean reconstruction error attained by CWF-SONAH is 15.5% at 500 Hz and 5.3% at 1000 Hz. Additionally, it is worth noting that the reconstruction errors obtained by PSWF-OMP exhibit slighter fluctuations across different sampling array configurations at both frequencies, while CWF-OMP and CWF-SONAH show dramatic fluctuations. This observation suggests that PSWF-OMP is more robust and less sensitive to variations in the sampling array randomness, further highlighting its reliability and consistent performance.

### 3.3. Effect of Sampling Numbers

In this subsection, we analyze the impact of the number of sampling points on the reconstruction results by CWF-SONAH, CWF-OMP and PSWF-OMP at frequencies of 500 Hz and 1000 Hz. The sampling numbers range from 50 to 625. For a fixed number of sampling points, 10 different configurations of the sampling array are evaluated. As a result, the mean reconstruction error can be derived for assessment. Additionally, error bars are used to evaluate the robustness of the reconstruction methods to different sampling array configurations.

As depicted in Figure 10, when the number of measurement points increases, the reconstruction errors obtained through the three algorithms exhibit a steady decrease trend. At the same time, the fluctuations in reconstruction errors become less prominent with increasing measurement points. Moreover, the error curves for PSWF-OMP, CWF-OMP and CWF-SONAH tend to converge at a high sampling number at both frequencies. This suggests that the sound field information may be recorded sufficiently at high sampling numbers, enabling a good approximation with a large number of wave functions. Therefore, the reconstruction accuracy can be enhanced and converge to the limit level. However, at low sampling numbers, the reconstruction errors produced by CWF-OMP and CWF-SONAH exhibit significant fluctuations and consistently remain at a high level. Specifically, the maximum reconstruction errors for CWF-OMP are nearly 30% at 500 Hz and approximately 35% at 1000 Hz. It seems that CWF-SOANH is rather sensitive to the measurement numbers since the maximum reconstruction errors are larger than 60% at both frequencies. In contrast, the mean errors obtained using PSWF-OMP remain below 10% across the entire range of sampling numbers at both frequencies. This implies that PSWF-OMP is less sensitive to the sampling numbers.

To explain the sensitivity of PSWF-OMP versus the sampling numbers, Figure 11 presents the coefficient sparsity changes in PSWF-OMP and CWF-OMP. In fact, sparsity refers to the number of wave functions employed to approximate the radiated sound field. It can be seen that the higher sampling numbers enable more cylindrical wave functions incorporated to model the radiated sound field, which can reduce the reconstruction errors at both frequencies (see Figure 9). This indicates that the performance of CWF-OMP is easily influenced by the sampling numbers, necessitating a greater number of wave functions to achieve higher reconstruction accuracy. In contrast, the coefficient sparsities in PSWF-OMP exhibit fewer changes with respect to sampling numbers, demonstrating a robust capability of PSWFs in representing the radiated sound field. In other words, the implementation of PSWF-OMP can significantly reduce the sampling requirements.

### 3.4. The Behavior across a Wide Frequency Range

Figure 12 illustrates the reconstruction errors obtained through CWF-SONAH, CWF-OMP and PSWF-OMP over a frequency range of 50~1000 Hz. The simulations utilized 50 sparse sampling points. To assess the impact of sampling array randomness, 10 different configurations of the sampling array were employed. The mean reconstruction errors and error bars were calculated to evaluate the performance of the reconstruction methods at each frequency. The results demonstrate that the mean reconstruction errors of PSWF-OMP consistently remain below 12% across the entire frequency range. Meanwhile, the mean reconstruction errors of CWF-OMP range from 10% to 28%, consistently higher than those of PSWF-OMP, except at 750 Hz. This exception may suggest that both PSWF-OMP and CWF-OMP exhibit similar representation sparsity for the interference pattern on the hologram surface at 750 Hz. The mean reconstruction errors of CWF-SONAH exhibit significant fluctuations, ranging from 5% to 23%. The comparison of the error bars indicates that the randomized sampling array has minimal impact on the reconstructed results of PSWF-OMP across a wide frequency range. Overall, the low level of mean reconstruction errors and the small error bars can provide strong evidence for the frequency robustness of the PSWF-OMP.

Time complexity is another important indicator for evaluating different reconstruction algorithms. The calculation time for 20 frequencies in Figure 12 using MATLAB 2022(a) was 1.41 s for CWF-SONAH, 0.97 s for CWF-OMP and 0.95 s for PSWF-OMP. The calculations were performed on a ThinkStation P300 manufactured by Lenovo in China, and only the time spent for regularizations and sound field reconstructions was included. Obviously, PSWF-OMP is the most efficient method. In other words, the proposed method has the best computational efficiency for the elongated source model.

## 4. Experiment

To further examine the feasibility of the proposed method, an experiment was conducted in an anechoic chamber. The experimental setup is depicted in Figure 13. Two Bluetooth loudspeakers with dimensions of 0.105×0.105×0.184 m were used as the sound sources. They were positioned at coordinates (0, 0, 0.175) m and (0, 0, −0.175) m to simulate a virtual elongated sound source extending along the *z*-direction. The origin of the Cartesian coordinate system was set at the midpoint between the two loudspeakers. Sinusoidal signals with frequencies ranging from 100 to 1000 Hz and an interval of 100 Hz were generated by a smartphone to drive the two loudspeakers (100 Hz is replaced by 150 Hz due to the mechanical resonance). Both loudspeakers were driven with the same amplitude and phase. The hologram surface and reconstruction surface were located at x=0.2 m and x=0.1 m, respectively. A planar array consisting of 8×8 microphones was used to measure the sound pressures on both the hologram surface and reconstruction surface. By sequential scanning, four patch measurements were integrated to expand the measuring area with the help of a fixed reference microphone. Finally, a total of 16×16=256 grid points were measured on the hologram surface and reconstruction surface with an identical dimension of 0.75×0.75 m^2^ and a uniform spatial interval of 0.05 m.

In our experiment, the hologram aperture was sufficiently large compared to the size of the virtual elongated sound source, and the microphone spacing was sufficiently small compared to half of the smallest wavelength across the frequency range of 100–1000 Hz. To validate the proposed method within a sparse framework, we randomly selected 80 sampling points from the hologram surface. The pressures at these sparse sampling points were utilized to reconstruct the pressures on the reconstruction surface. The measured pressures on the reconstruction surface are regarded as benchmark values. By comparing the reconstructed pressures with the benchmark values, we were able to evaluate the performance of different reconstruction methods.

Figure 14 and Figure 15 compare the reconstructed pressures obtained from CWF-SONAH, CWF-OMP and PSWF-OMP with the benchmark values at 500 Hz and 1000 Hz on the reconstruction surface. It is clear that both CWF-SONAH and PSWF-OMP exhibit similar reconstruction patterns to the benchmark pressures at 500 Hz. Moreover, the reconstructed patterns appear smoother than the measured pressure in Figure 14d, indicating a noise suppression effect in PSWF-OMP. Upon evaluating the reconstruction errors at 500 Hz, we find a 21.2% error for PSWF-OMP, a 17.5% error for CWF-SONAH and a 26.5% error for CWF-OMP. However, at 1000 Hz, the reconstruction errors significantly increase to 33.4% for PSWF-OMP, 41.6% for CWF-SONAH and 48.7% for CWF-OMP. Notably, the increase in error for PSWF-OMP is smaller compared to those of CWF-OMP and CWF-SONAH. Furthermore, the reconstructed patterns of CWF-OMP and CWF-SONAH at 1000 Hz display noticeable distortions near the edges, suggesting that utilizing only 80 sampling points may not be sufficient. Overall, PSWF-OMP demonstrates low-level reconstruction errors at both frequencies, indicating the superior performance of the proposed method compared to CWF-OMP and CWF-SONAH when the number of sampling points is limited.

Figure 16 displays the reconstruction errors of CWF-SONAH, CWF-OMP and PSWF-OMP for different sampling numbers. The mean reconstruction errors and error bars are calculated based on ten randomly selected sparse arrays with the same sampling number. It is evident that the reconstruction errors decrease as the sampling number increases for both methods at 500 Hz and 1000 Hz. Across the entire range of sampling numbers, PSWF-OMP consistently exhibits lower mean reconstruction errors and smaller error bars compared to CWF-OMP. This indicates that PSWF-OMP is less sensitive to changes in sampling numbers. Though CWF-SONAH outperforms PSWF-OMP in terms of reconstruction accuracy at several intermediate sampling numbers at 500 Hz, the reconstruction errors escalate significantly when the number of measurements decreases below 50. Figure 17 provides the coefficient sparsities of PSWF-OMP and CWF-OMP for different sampling numbers. Consistent with the simulation results, the coefficient sparsities in PSWF-OMP show minimal changes with respect to the sampling numbers at 500 Hz, highlighting its lower requirement for sampling numbers. However, at 1000 Hz, there is an observable increasing trend in the coefficient sparsities of PSWF-OMP, although they remain consistently lower than those of CWF-OMP. This phenomenon may be attributed to the complex sound field structure generated by the two loudspeakers. As a result, more wave functions are needed to approximate the radiated sound field. Overall, PSWF-OMP demonstrates greater robustness in reconstructing the radiated sound field compared to CWF-OMP, as evidenced by its lower reconstruction errors and smaller sensitivity to sampling numbers.

Figure 18 illustrates the reconstruction errors of CWF-SONAH, CWF-OMP and PSWF-OMP versus frequency with 80 sampling points. It is evident that the reconstruction errors of PSWF-OMP remain below 30% in the frequency range of 100 to 700 Hz, which is lower than those of CWF-OMP and CWF-SONAH. However, when the frequencies exceed 800 Hz, the differences in error between CWF-OMP and PSWF-OMP become less prominent, and both methods achieve comparable results. The reason for this observation may lie in the fact that using only two loudspeakers is not sufficient to construct a featured elongated sound source model, especially at high frequencies. As a result, the advantages of PSWF-OMP for elongated source models cannot be fully displayed. At high frequencies, the radiated sound field has a strong interference effect, requiring higher wavenumber components of both PSWFs and CWFs to approximate the sound field (as shown in Figure 17b). In such cases, the superior matching performance of PSWFs for elongated sound source models is not prominently demonstrated. Overall, the PSWF-OMP demonstrates better reconstruction results across the entire frequency range, particularly at low frequencies.

## 5. Conclusions

A sound field reconstruction method based on prolate spheroidal wave superposition and sparse regularization is suggested to reconstruct the radiated sound field of an elongated sound source model. The proposed method represents the radiated sound field as a linear superposition of the prolate spheroidal wave functions (PSWFs) and uses the orthogonal matching pursuit (OMP) algorithm to find the sparse coefficient solutions via a small number of measurements. In the numerical simulations, A comparison is conducted between the proposed method and the traditional SONAH algorithm, emphasizing the advantages of the former in handling elongated sound sources when the number of measurements is insufficient. Moreover, the proposed method is compared to a method that integrates cylindrical wave representation with the OMP algorithm to illustrate the compatibility of PSWFs with elongated sound sources. The results show that the proposed method can realize better reconstruction accuracy with a reduced number of measurement points. The robustness of the proposed method is validated by the smaller reconstruction errors with respect to a wide frequency range, different array configurations and a broad range of sampling numbers. The experimental results align with the simulation results, further validating the feasibility and reliability of the method.

The proposed method still has limitations regarding the calculation of PSWFs. This calculation is dependent on the parameter c=kd/2=πd/λ. Furthermore, when the interfocal distance d is set too large or the frequency is too high, the calculation precision of PSWFs may decrease, thereby affecting the accuracy of the reconstructed sound field.

In future studies, the application of PSWFs will be extended to more complex elongated source models, such as real unmanned underwater vehicles. Furthermore, alternative sparse regularization techniques will be explored and integrated with PSWFs to assess their performance. In addition, the configurations of PSWFs, such as the interfocal distance, will be further investigated to match the sources. A suitable geometry configuration of the hologram surface and the arrangement of measurement points will also be explored.

## Figures and Tables

**Figure 1 sensors-23-08312-f001:**
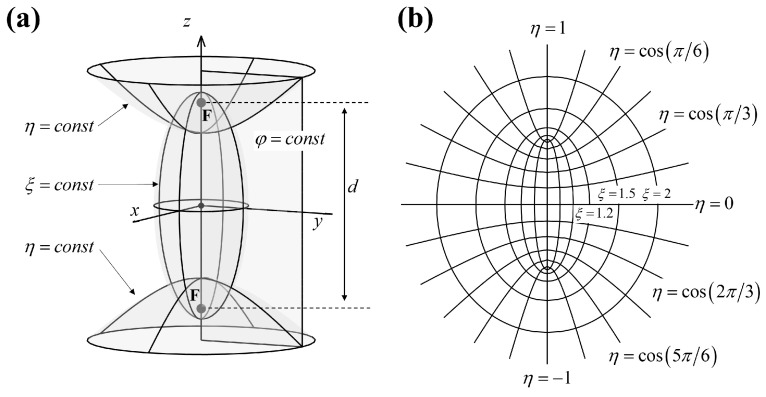
Prolate spheroidal coordinate system: (**a**) coordinate surfaces and (**b**) confocal ellipses and hyperbolas.

**Figure 2 sensors-23-08312-f002:**
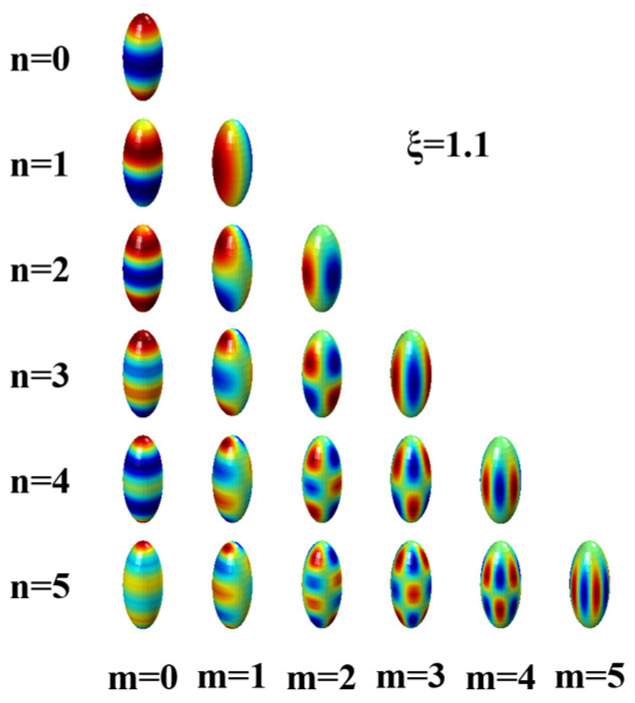
Real parts of the first few outgoing prolate spheroidal harmonics mapped to a prolate spheroid with respect to ξ=1.1. The red/blue map represents positive/negative values.

**Figure 3 sensors-23-08312-f003:**
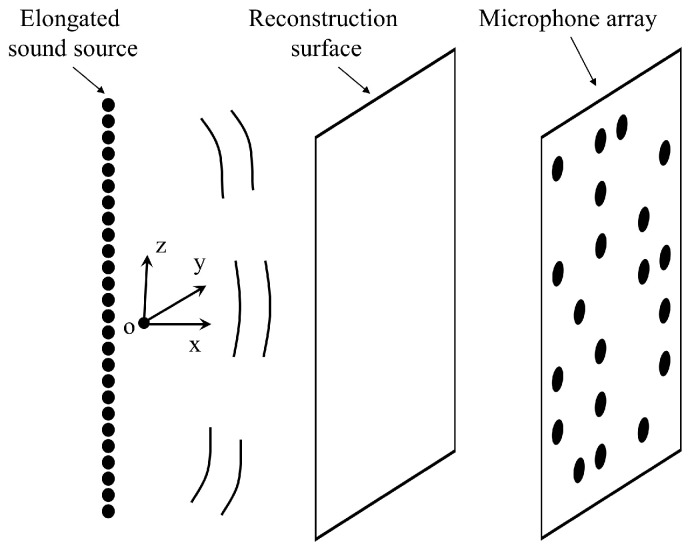
Diagram of the sound field reconstruction for an elongated sound source.

**Figure 4 sensors-23-08312-f004:**
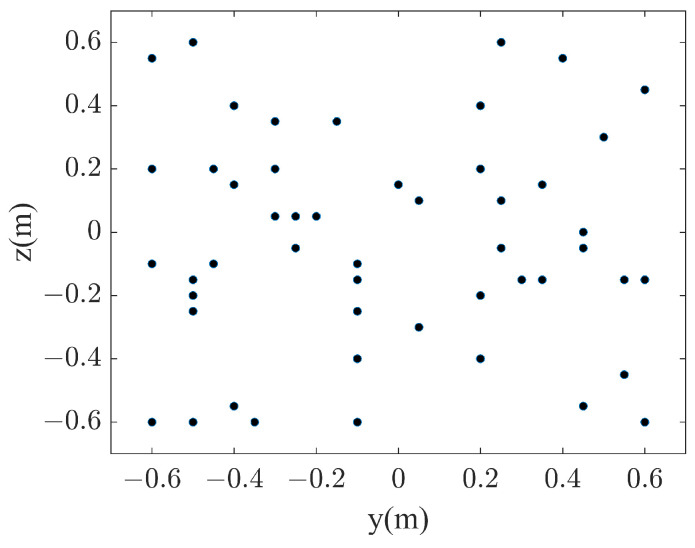
The microphone array consists of 50 elements that are randomly distributed.

**Figure 5 sensors-23-08312-f005:**
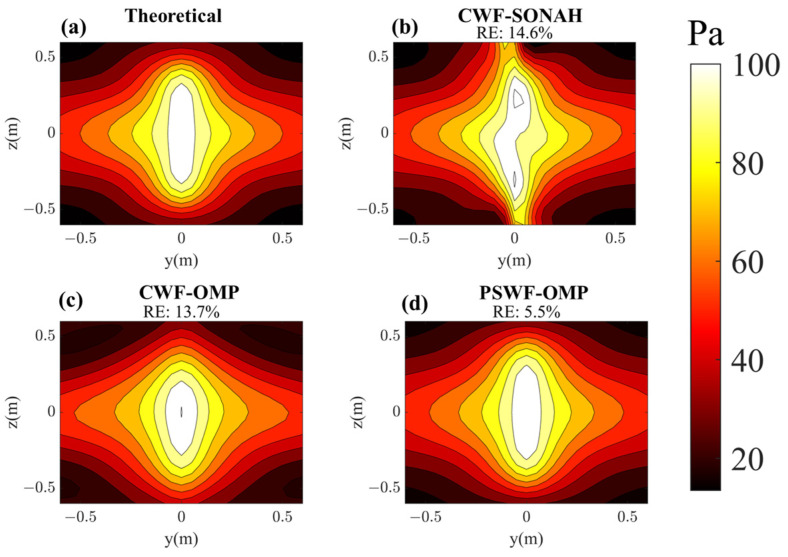
Pressure on the reconstruction surface with respect to line-array monopoles at 500 Hz. (**a**) Theoretical; (**b**) reconstructed by CWF-SONAH; (**c**) reconstructed by CWF-OMP; (**d**) reconstructed by PSWF-OMP.

**Figure 6 sensors-23-08312-f006:**
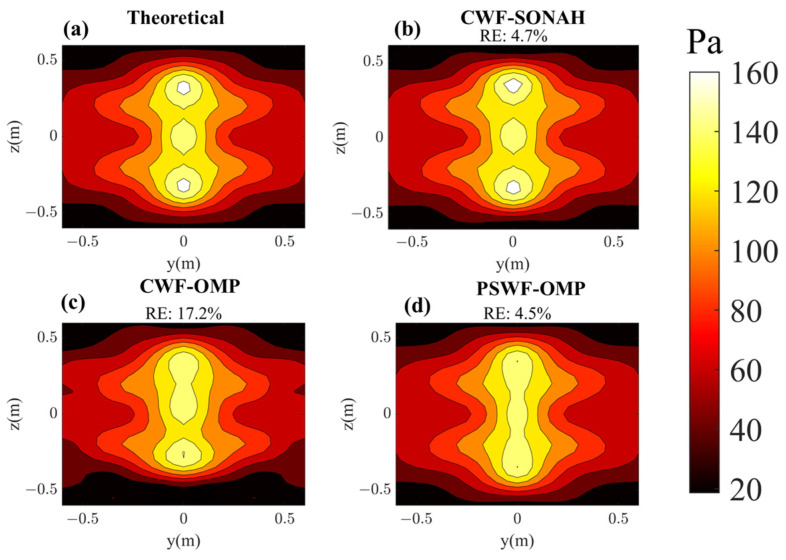
Pressure on the reconstruction surface with respect to line-array monopoles at 1000 Hz. (**a**) Theoretical; (**b**) reconstructed by CWF-SONAH; (**c**) reconstructed by CWF-OMP; (**d**) reconstructed by PSWF-OMP.

**Figure 7 sensors-23-08312-f007:**
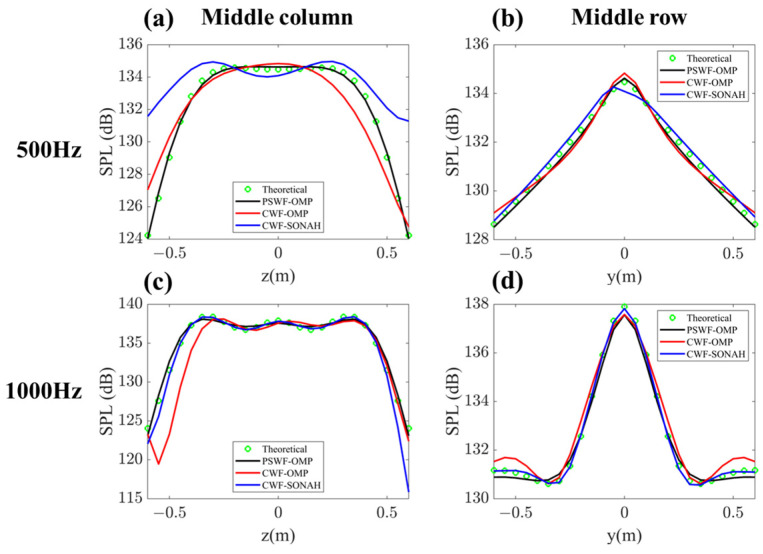
A comparison of the reconstructed pressures by CWF-SONAH, CWF-OMP and PSWF-OMP with the theoretical pressures along the middle column line and middle row line on the reconstruction surface at 500 Hz and 1000 Hz.

**Figure 8 sensors-23-08312-f008:**
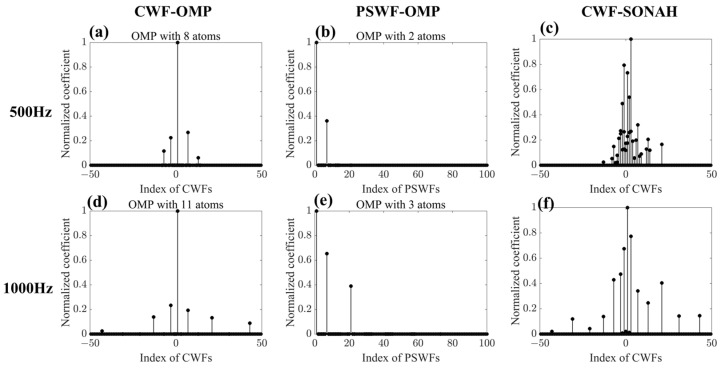
A comparison of the normalized coefficients in CWF-SONAH, CWF-OMP and PSWF-OMP at 500 Hz and 1000 Hz.

**Figure 9 sensors-23-08312-f009:**
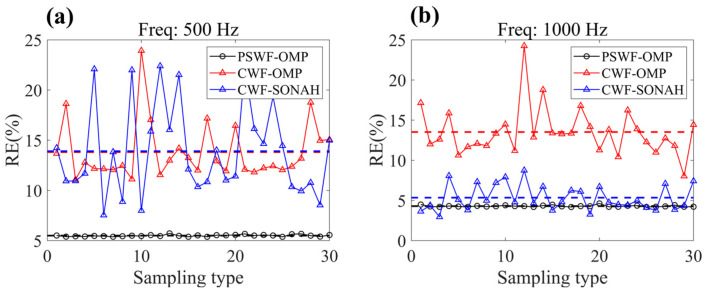
Influence of the randomness of the sampling array. (**a**) 500 Hz, (**b**) 1000 Hz.

**Figure 10 sensors-23-08312-f010:**
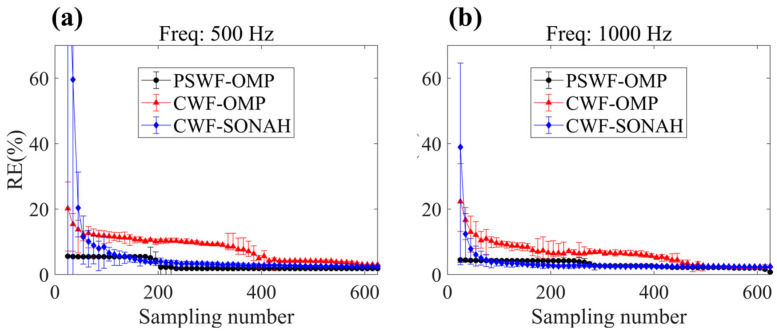
Reconstruction errors of CWF-SONAH, CWF-OMP and PSWF-OMP with respect to line-array monopoles under different sampling numbers. (**a**) 500 Hz, (**b**) 1000 Hz.

**Figure 11 sensors-23-08312-f011:**
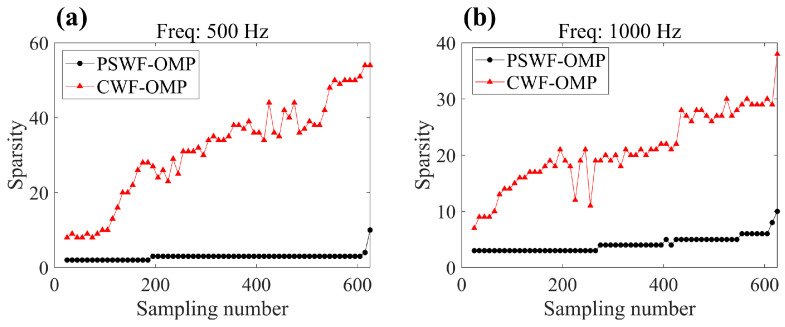
The sparsity of PSWF-OMP and CWF-OMP with respect to line-array monopoles under different sampling numbers. (**a**) 500 Hz, (**b**) 1000 Hz.

**Figure 12 sensors-23-08312-f012:**
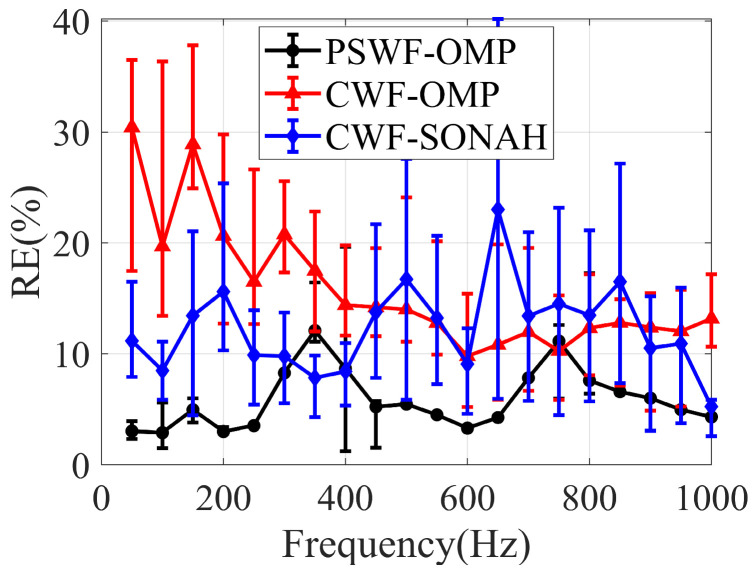
Reconstruction errors of CWF-SONAH, CWF-OMP and PSWF-OMP versus frequency with 50 sampling points.

**Figure 13 sensors-23-08312-f013:**
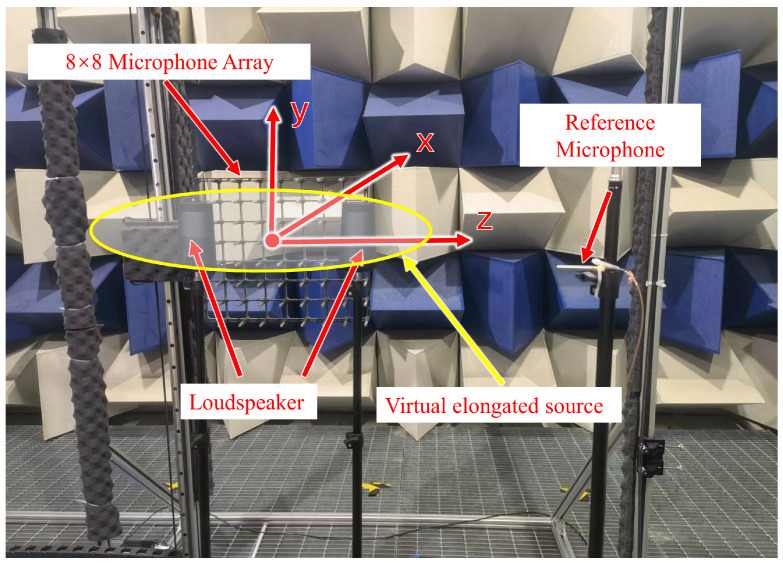
The experimental setup.

**Figure 14 sensors-23-08312-f014:**
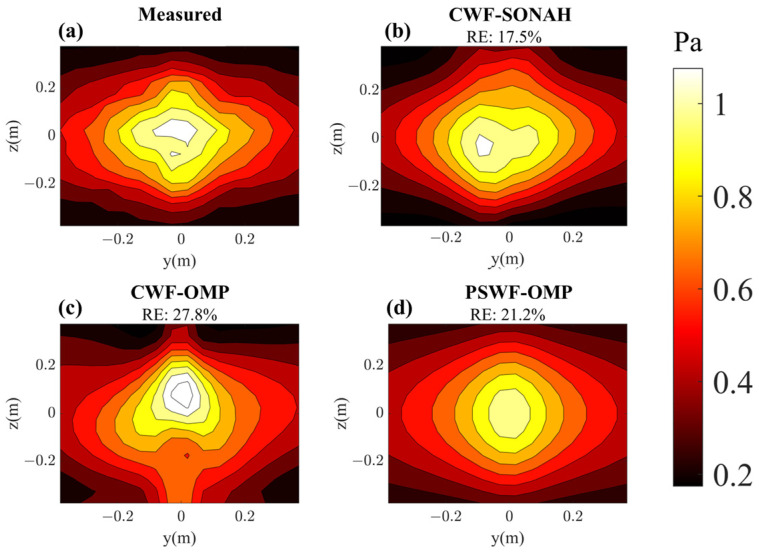
Pressure on the reconstruction surface with respect to two loudspeakers at 500 Hz. (**a**) Theoretical; (**b**) reconstructed by CWF-SONAH; (**c**) reconstructed by CWF-OMP; (**d**) reconstructed by PSWF-OMP.

**Figure 15 sensors-23-08312-f015:**
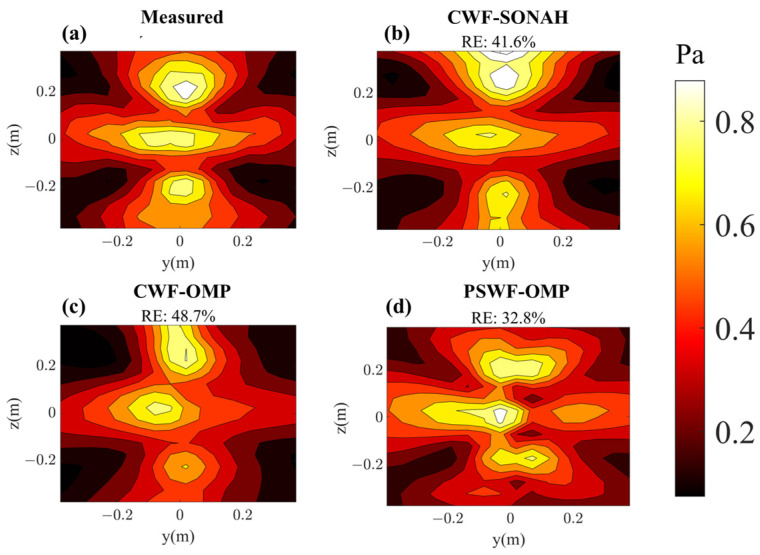
Pressure on the reconstruction surface with respect to two loudspeakers at 1000 Hz. (**a**) Theoretical; (**b**) reconstructed by CWF-SONAH; (**c**) reconstructed by CWF-OMP; (**d**) reconstructed by PSWF-OMP.

**Figure 16 sensors-23-08312-f016:**
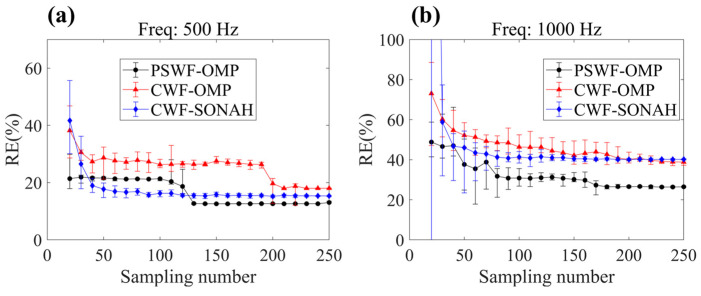
Reconstruction errors of CWF-SONAH, CWF-OMP and PSWF-OMP with respect to two loudspeakers under different sampling numbers. (**a**) 500 Hz, (**b**) 1000 Hz.

**Figure 17 sensors-23-08312-f017:**
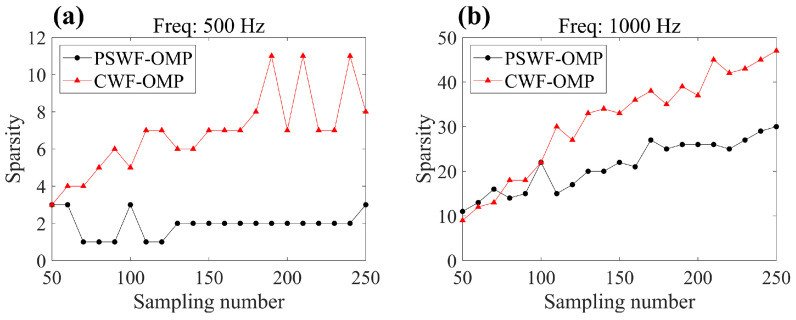
The sparsity of PSWF-OMP and CWF-OMP with respect to two loudspeakers under different sampling numbers. (**a**) 500 Hz, (**b**) 1000 Hz.

**Figure 18 sensors-23-08312-f018:**
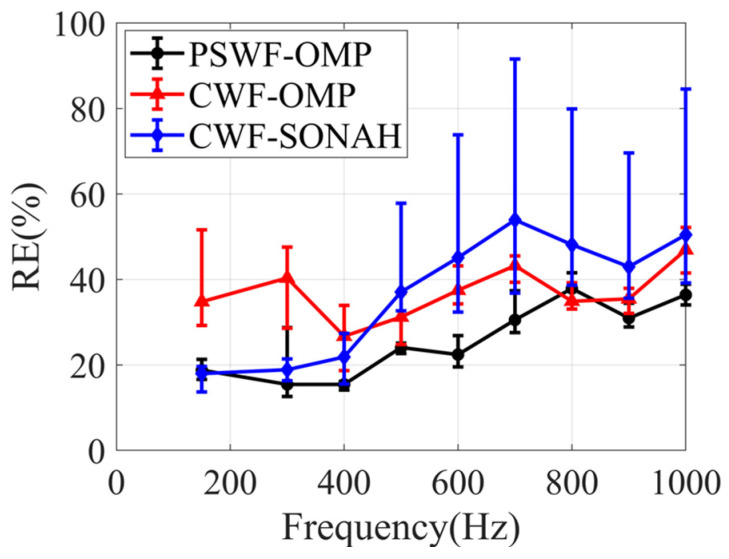
Reconstruction errors of CWF-SONAH, CWF-OMP and PSWF-OMP versus frequency with 80 sampling points.

## Data Availability

Not applicable.

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
