# Peer review of "Sound Field Reconstruction Using Prolate Spheroidal Wave Functions and Sparse Regularization"

_sensors, 2023, doi:10.3390/s23198312_

Round 1

Reviewer 1 Report

The paper is based on the reconstruction of sound field reconstruction using prolate spheroidal wave functions (PSWFs) and the well-known OMP algorithm in sparse signal processing.

The paper is very well written and clearly explained. However, few concerns are here:

1.     The novelty is not clear to me, especially without any introduction to the work which has been previously done (or lack of) in this specific area.

2.     The authors use the PSWF to represent the basis function for further analysis. The authors do not explain how do these basis function meet the condition of using compressive sensing and sparse signal reconstruction methods, which may not be the case at all. 

3.     There is no comparison of the reconstruction performance with some other, traditional ways of the reconstruction (the authors state “The performance of the proposed method is compared with traditional NAH algorithms that employ cylindrical wave functions to represent the radiated sound field of elongated sources”, but the “traditional NAH algorithm” is still OMP, just with a different basis function), which leads me to point 4.

4.     If there is no clear reason or advantage for the reconstruction of such signals, point 1 should be better explained.

The use of English language of this paper is good. Few typos and mistakes which, by careful double-check, can be improved. 

One of the suggestions to the authors is not to start a sentence with "And..."

Reviewer 2 Report

Contributions must be clarified.

Insert methodology (methods and materials) section. Actually, it is presented with numerical results.

Comparison with more methods, some presented in the introduction, should be considered along with the performance results.

Comparison in terms of complexity must be added.

Future works with details should be presented in the conclusion section.

At the end of the paper, abbreviations must be inserted. Please refer to the MDPI template.

Reviewer 3 Report

This paper presents a sound field reconstruction method based on prolate spheroidal wave functions (PSWFs) and the orthogonal matching pursuit (OMP) algorithm. Detailed comments and suggestions are given below.

1) It would be interesting to explain the reason why using prolate spheroidal wave functions instead of spheroidal wave functions.

2) It would be better to check whether the present method produces an ill-conditioning matrix and leads to numerical instability with the increasing truncated term.

3) It would be better to mention the limitation of the present study in the conclusion part.

Minor editing of English language is required

Round 2

Reviewer 2 Report

The paper was improved and extended, congrats.